# Knowledge, Attitude and Practices towards the Prevention of Schistosomiasis Mansoni in an Endemic Area of Alagoas, Northeast Brazil

**DOI:** 10.3390/tropicalmed8010034

**Published:** 2023-01-03

**Authors:** Adriano José dos Santos, Shirley Verônica Melo Almeida Lima, Alvaro Francisco Lopes de Sousa, Aytana Vasconcelos dos Santos, Israel Gomes de Amorim Santos, Márcio Bezerra Santos, Vera Lucia Corrêa Feitosa, Allan Dantas dos Santos, Juliana Cristina Magnani Primão, Denise de Andrade, José Rodrigo Santos Silva

**Affiliations:** 1Post-Graduation Programme in Parasitic Biology, Federal University of Sergipe, São Cristóvão 49100-000, SE, Brazil; 2Post-Graduation Programme in Nursing, Federal University of Sergipe, São Cristóvão 49100-000, SE, Brazil; 3Collective Health Research Center, Federal University of Sergipe, São Cristóvão 49100-000, SE, Brazil; 4Global Health and Tropical Medicine, Instituto de Higiene e Medicina Tropical, Universidade Nova de Lisboa, 1349-008 Lisbon, Portugal; 5Instituto de Ensino e Pesquisa, Hospital Sírio-Libanês, São Paulo 01308-060, SP, Brazil; 6Post-Graduation Programme in Pharmaceutical Sciences, Federal University of Sergipe, São Cristóvão 49100-000, SE, Brazil; 7Biology Department, State University of Alagoas, Campus II, Santana do Ipanema 57500-000, AL, Brazil; 8Medical and Nursing Science Center, Federal University of Alagoas, Campus Arapiraca 57309-005, AL, Brazil; 9Post-Graduation Programme in Health Sciences, Federal University of Sergipe, São Cristóvão 49100-000, SE, Brazil; 10Department of Morphology, Federal University of Sergipe, São Cristóvão 49100-000, SE, Brazil; 11Human Exposome and Infectious Diseases Network (HEID), Ribeirão Preto College of Nursing, University of São Paulo, Ribeirao Preto 14040-902, SP, Brazil; 12Department of Statistics and Actuarial Sciences, Federal University of Sergipe, São Cristóvão 49100-000, SE, Brazil

**Keywords:** endemic diseases, knowledge, practices, attitude, neglected tropical diseases, health education, prevention, public health, *Schistosoma mansoni*

## Abstract

We analyzed the knowledge, attitudes and practices (KAP) of schistosomiasis mansoni prevention in an endemic area of Brazil. This cross-sectional study was conducted between March and May 2021, with 412 participants living in the municipality of Feira Grande, Alagoas, Brazil. Data collection occurred through visits to the Health Center Urbano II and Massapê, through an interview with a structured questionnaire to identify the levels of KAP regarding schistosomiasis prevention. Of all respondents, 70.87% lived in rural areas, 22.66% reported a history of past schistosomiasis and 52.71% never participated in schistosomiasis control program actions. Factors associated with better KAP scores were being part of an older age group, not using rainwater and having no history of past schistosomiasis. Specifically, among the domains, attitude was the highest score and knowledge was the lowest. Participation in a health intervention program, knowing someone who had schistosomiasis and having been informed through a public health program seemed to have an important impact on the population’s KAP. Our results contributed to broadening perceptions about schistosomiasis prevention, highlighting the positive impacts that health programs and interventions have on disease control.

## 1. Introduction

Schistosomiasis affects approximately 240 million people worldwide [1] and is still considered a public health problem. Within the group of neglected tropical diseases (NTDs), Schistosomiasis is one of the most common parasitic diseases in the world and one of the most important waterborne diseases in America [2,3], causing a significant economic and health impact on the population. Its symptoms range from fever and fatigue to diarrheal outbreaks, low cognitive development in children, work disability in adults, hepatosplenomegaly, digestive bleeding and ascites in the most severe untreated cases, which can culminate in death [4,5]. 

Prevalent in tropical and subtropical areas, schistosomiasis mainly affects the poorest and most rural communities, where access to potable water and basic sanitation is poor. This population group is subsidized by agricultural and fishing activities, as well as for the habit of performing domestic and recreational activities in natural water collections [1,6,7]. Besides the aforementioned context, sociocultural aspects, associated with biogeographical conditions, unhealthy hygienic habits and a low level of population knowledge, corroborate its occurrence [8,9,10].

Schistosomiasis mansoni is a parasitic illness caused by the helminth *Schistosoma mansoni*, endemic in 51 countries [1] including Brazil, with a wide geographic distribution in the country and records of cases in 19 of its 27 federated units, highlighting the northeast and southeast Brazilian regions [11]. These are at opposite ends of the Municipal Human Development Index (HDI), with the worst (0.663) and best (0.766) indexes, respectively, among the five Brazilian macroregions [11], with an estimated 1.5 million people living at risk of infection in endemic regions [12,13]. In the period from 2009 to 2019, 423,117 (4.9%) people infected with *S. mansoni* were identified out of a total of 9,867,120 people examined in the Brazilian territory [14]. In the municipality of Feira Grande, Alagoas, 437 tests were performed for the diagnosis of schistosomiasis in 2021, with a positivity rate of 1.83% (8). It is important to emphasize that this is an area considered endemic for schistosomiasis in the country [15]. 

The high positivity added to the limitations of preventive drug therapies can lead to constant reinfection in endemic areas [5,16], associated with communities’ risk habits, sometimes motivated by little sense of self-care as a result of low knowledge about disease prevention [17,18]. In this perspective, studies that address knowledge, attitudes and practices (KAP) are based on the hypothesis that good health behavior comes from obtaining adequate knowledge, which reflects in positive attitudes and, consequently, in good health practices [19,20]. Thus, community mobilization through health education will become indispensable for raising the communities’ awareness about schistosomiasis [9,21].

In this context, it becomes relevant to investigate the perceptions of knowledge, attitudes and practices of the population regarding schistosomiasis. This type of study can reveal limitations, gaps in knowledge, cultural beliefs and/or behaviors unknown to health agencies, which interfere in the effectiveness of control measures, also revealing the social context in which the disease is inserted and opening paths for control measures relevant to the reality of the population [22]. Therefore, the present study aimed to analyze the knowledge, attitudes and practices of schistosomiasis prevention in a population residing in an endemic area in Brazil. 

## 2. Materials and Methods

### 2.1. Type of Study

This was an observational, cross-sectional situational diagnosis study with a quantitative approach.

### 2.2. Study Area

The study area corresponded to the municipality of Feira Grande (Appendix A), located in the Agreste mesoregion of Alagoas state, Brazil. This municipality was part of the seventh (7th) Health Region (SR) of Alagoas, which was the second most populous region and had 9 municipalities endemic for schistosomiasis [23].

Feira Grande (09°54′01″ S; 36°40′39″ W) had a territorial area of 175,906 km^2^, a tropical rainy climate with a dry summer and a rainy season in the fall/winter. It had an estimated population of 22,192 inhabitants in 2021, with 3421 (16.05%) living in urban and 17,900 (83.95%) in rural areas, with a demographic density of 123.42 inhabitants/km^2^, an HDI of 0.533 (medium) and a GINI of 0.535 [24,25,26].

According to data from the last census regarding sanitation, it was estimated that 7.8% of households in the municipality had adequate sanitary sewage and 3% of urban households were on public roads with adequate urbanization. Furthermore, about 2400 houses are in a situation of active connection, being supplied by the water supply service [24,25]. 

The choice of Feira Grande, Alagoas, Brazil, as the study area considered the following criteria: the municipality was part of the endemic area for schistosomiasis in the state of Alagoas and had a history of high prevalence rates; the identification of an intermediate host breeding site in the territory; the socioeconomic conditions of the population, with the practice of subsistence agriculture using water from natural sources; and the scarcity of investigative research (academic production) related to the KAP in the area.

### 2.3. Study Population and Sampling

Samples were composed of individuals who lived in the municipality investigated, and who received care from health services in the Health Center (HC) Urban II, located in downtown Feira Grande, and in the rural area of Massapê District, located about 5.7 km from the city center (Appendix A). The choice of these HCs considered the registration of schistosomiasis cases according to the Municipal Health Department. 

According to information provided by the Municipal Health Department of Feira Grande, the HC Urbano II was divided into 9 microareas and served a total of 1282 families, whereas the HC Massapê district was divided into 7 microareas and served 960 families. Thus, the study universe was composed of 2242 families and approximately 8968 individuals. 

Sample size calculation was performed using the formula of qualitative variables with a finite population [27], admitting no previous knowledge about KAP in the study population (p=0.5 e q=0.5), a population of 8968 individuals (N), with a significance level of 95% (Z(α/2) = 1.96) and a margin error of 5% (ε=0.05). The minimum sample size calculated was 368 individuals.

Participation in the study was voluntary, and all individuals included in the study met the following criteria: living in one of the locations served by the aforementioned HC; being at least 15 years old, assuming they had a level of education that allowed them to obtain information about schistosomiasis; and signing the informed consent form (ICF) for those over 18 years old, and the informed assent form (IAF) for minors, where their parents or guardians consented to their participation. Subsequently, the study excluded individuals who resided in locations outside the coverage area of the aforementioned HC; who were younger than the established minimum age limit; or those who, even fulfilling the previous criteria, refused to sign the ICF. 

### 2.4. Data Collection

Data collection occurred between March and May 2021 through visits to the referred HC. The collection procedure occurred in two ways: I—spontaneous demand in the HC, where participants were approached during routine care in the units and invited to participate in the study; II—active search in residences, where visits were performed by community health agents (CHA) who indicated the habitations to be visited. 

Individual interviews were conducted using a structured questionnaire, and the instrument contained questions related to the population’s knowledge, attitudes and practices regarding schistosomiasis, as well as questions aimed at identifying the participants’ sociodemographic profile and health service-seeking behavior. This instrument was developed by the authors of this study through bibliographic research and a survey of studies that investigated the aspects of KAP related to schistosomiasis, which were the basis for the questionnaire’s construction [10,18,22,28,29].

In order to minimize understanding problems during the interview, the questionnaire was applied considering all self-declared answers of the interviewees, and the answers regarding KAP were quantitatively evaluated. 

### 2.5. Variables of Study

In this study, the aspects of knowledge (K), attitudes (A) and practices (P) were assessed through structured questionnaires with 12, 6 and 6 questions, respectively. Each item in the questionnaires had a minimum score of 0 points and a maximum score of 1 point. In items with more than one correct alternative, scores were considered proportional to the number of correct answers, considering equal weights for the alternatives. The questionnaire underwent a validation process by experts regarding objectivity, clarity and relevance. The content validity coefficient (CVC) was calculated to obtain consensus. 

Final KAP scores were defined as the arithmetic mean of the scores within each domain. In this way, the scores were represented by quantitative variables (dependent), described within the closed interval between 0 and 1, where the higher the score value, the higher the evaluation of the respondent’s KAP. These same scores were correlated with the variables sociodemographic character (independent) and search for health service and care (independent). 

It is noteworthy that, although question 12 (source of knowledge about the disease) was included in the knowledge sub-questionnaire, it was grouped as an independent variable in the statistical analyses, together with the variables of the sociodemographic sub-questionnaire; therefore, in order to calculate the knowledge score, 11 questions were considered. 

### 2.6. Statistical Analysis

Data were stored in a Microsoft Excel 2016 database (Microsoft Corporation; Redmond, WA, EUA) where scores of each component were calculated. Later, statistical analyses were performed in R 4.0.0 software (RStudio 1.4.1717 (GNU Affero General Public License). Descriptive analyses of the independent variables were performed, calculating absolute and relative (%) frequencies. For the dependent variables in the descriptive analysis, median, interquartile interval, and minimum and maximum values were calculated. Adherence to the normal distribution of the scores was checked using the Kolmogorov–Smirnov test [30]. 

Associations of the scores with independent variables were performed through the nonparametric Mann–Whitney and Kruskal–Wallis tests [30], with the respective measures of effect size: r statistic [31], in which values between 0.10 and 0.3 were classified as a small effect, between 0.30 and 0.5 were classified as a moderate effect and greater than or equal to 0.5 were classified as a large effect; and the Eta-Square statistic [31], where values between 0.01 and 0.06 were classified as a small effect, between 0.06 and 0.14 were a moderate effect, and greater than or equal to 0.14 were a large effect. To evaluate the relationship between the scores, Spearman correlation [30] was applied and a Kruskal–Wallis test [30] was applied to compare the scores.

For multivariate analysis, independent variables that showed no significant relationship with at least one score were initially discarded. The remaining variables were dichotomized and then transformed using factor analysis [32,33], in order to solve multicollinearity problems. In the factor analysis, the tetrachoric correlation matrix was adopted as a measure of association. The number of factors adopted was based on Kaiser’s criterion, in which factors where the eigenvalue equal to or greater than 1 were adopted. For better visualization of the factor weights, the varimax rotation matrix was adopted. The beta regression model [34,35] was used to verify the multiple associations of the KAP scores with the independent variables, represented through the factor analysis factors. 

### 2.7. Ethical Considerations

The present study was developed according to the ethical precepts of Resolution 466/12 of the National Health Council and its complementary resolution, being approved by the Ethics Committee for Research Involving Humans from the Federal University of Sergipe under opinion number 4442234 (CAAE: 37293020.4.0000.5546).

## 3. Results

### 3.1. Profile of the Interviewees

We interviewed 428 individuals, from which 16 (3.74%) were excluded according to the exclusion criteria, thus composing a sample of 412 participants. Regarding socio-demographic characteristics, it was observed that 70.87% (n = 192) of the interviewees lived in rural areas, 74.27% (n = 306) were female, 44.17% (n = 182) were in the young adult age group (20 to 39 years old), and 40.53% (n = 167) had an illiterate or incomplete elementary school education. When asked about their economic status, 65.29% (n = 269) of the participants stated that they were not economically active, and 65.05% (n = 268) were engaged in agriculture. Regarding the water supply and sanitation infrastructure, 71.12% (n = 293) had their houses supplied with piped water by the Alagoas sanitation company, and 99.76% (n = 411) had a functional bathroom in their residence (Table 1).

About the service and health care demand profile, it was highlighted that 82.52% (n = 340) of the interviewees only look for HC services when needed, 22.66% (n = 80) had a self-reported history of positive infection by *S. mansoni*, 52.71% (n = 204) had never participated in Schistosomiasis control program (SCP) actions and 56.93% (n = 234) knew someone who had already had schistosomiasis. Regarding the first source of knowledge about the disease, public health services stand out (32.52%; n = 134), acquaintances and/or neighbors being 22.57% (n = 93) and relatives and/or older people being 22.33% (n = 92) (Table 1).

### 3.2. Analysis of KAP Scores 

Analyzing the KAP scores, it was observed that the attitude variable obtained the highest value, whereas knowledge had the lowest score. When comparing the knowledge and practice scores, the two variables had a significant difference (*p* < 0.001) between them, with good practice scores being higher than knowledge (Table 2). 

The descriptive analysis of knowledge, attitude and practice scores showed that among the three components, the attitude score (Md = 0.83) was higher than the practice (Md = 0.67) and knowledge scores (Md = 0.66). Overall, despite the small difference between these last two scores, the attitude variable had a significant difference compared to the knowledge (*p* < 0.000) and practice (*p* < 0.000) variables, presenting a higher score, thus resulting in a lower level of knowledge when compared to the other variables (Table 2).

The Spearman correlation between knowledge and attitudes (r = 0.351; *p* < 0.000) was positive and significant, indicating that the higher the knowledge score, the higher the attitude score. The correlation coefficient between knowledge and practices (r = 0.301; *p* < 0.000) was also positive and significant, showing that the higher the knowledge score, the better the practice conduct. Regarding the correlation between attitudes and practices (r = 0.292; *p* < 0.000), it was positive and significant, pointing out that the higher the attitude score, the better the practice score (Table 2).

### 3.3. Association of KAP Scores with Socio-Demographic Profile

Regarding the knowledge component, it was observed that older individuals had a better knowledge score on schistosomiasis, as well as those with a higher level of education, those who are economically active, and those who declared having another occupation, except for being a farmer. In addition, participants who had their houses supplied by the utility company and did not use rainwater also know more about the disease (Table 3). 

Regarding the attitudes component, it was found that respondents in older age groups had better attitudes about the disease, as do those who did not use rainwater as a supply source. Regarding the practices component, it was verified that rural area residents had better practices when compared to other areas of residence. The female interviewees had better practices than the male ones, as do older people when compared to younger ones. The interviewees who did not have piped water and transported it by tanker truck had better practices than other sources of supply (Table 3).

The results referring to the attitude scores regarding the sources of water supply, through piped and well water, had equal medians in the classes that compose these variables. Thus, these results were considered inconclusive, even in the presence of statistical significance. The knowledge and attitudes variables had no statistically significant association with the area of residence and interviewees’ gender (Table 3). 

In addition, sources of water supply from wells, dugouts and tankers had no significant association with knowledge, as well as attitudes had no association with these last two variables, nor practices with the first two. Still regarding attitudes and practices, there was no significant association with education, economic status and occupation (Table 3).

### 3.4. Association of KAP Scores with the Profile of Health Care and Services Demand

People with a positive history of past schistosomiasis, as well as those who had participated in some actions of a disease control program, had better levels of knowledge. Those who received information about parasitosis through public health services and through doctors or at the hospital were better informed (Table 4). 

Interviewees who had had schistosomiasis presented a more satisfactory level of attitudes than the others. Furthermore, those who had already had schistosomiasis, those who participated in SCP actions, those who knew someone who had already had the disease and those who were informed about it through a public health program had daily practices (Table 4).

When considering the attitude score, it was observed that variables such as participation in SCP actions, knowing someone who had already had schistosomiasis and having a public health program as a source of information had equal medians in the classes that compose these variables; thus, these results were considered inconclusive. The same applies to the practice score when considering the frequency variable with which they used health services in HC (Table 4).

The frequency with which interviewees used health services in HC was not statistically significant when associated with knowledge and attitudes. Regarding the source of knowledge about the disease, school, acquaintances and/or neighbors and relatives and/or older people had no significant association with any of the KAP variables, and being informed about schistosomiasis by a doctor or at the hospital was not statistically significant with attitude and practice scores (Table 4).

### 3.5. Multivariate Association of KAP Scores with Sociodemographic Profile and the Profile of Health Care and Services Demand

When performing factor analysis, the dimension of independent variables was reduced from 20 to 8 variables, keeping 80.5% of the information. The analysis quality was checked with Bartlett’s test (*p*-value < 0.001) and the communality values (greater than 0.70). The results presented the following factor structure (Appendix A):

Factor 1: people who lived in rural areas, were elderly, farmers, with a low level of education or did not have school as a source of information on schistosomiasis;

Factor 2: people who did not have piped water and used water that came from the rain and was transported by tanker truck;

Factor 3: people who had already participated in some health intervention program, who knew someone who had schistosomiasis and who had some source of information about schistosomiasis, in this case, a public health program;

Factor 4: people who also did not have piped water, used well water and had acquaintances or neighbors as a source of knowledge about this parasitosis;

Factor 5: formed by people whose source of knowledge about schistosomiasis was a doctor or at the hospital;

Factor 6: female individuals who self-report not being economically active;

Factor 7: people who used health services of the basic health units and who had never had schistosomiasis;

Factor 8: individuals who learned about the disease from relatives and/or older people.

A significant negative association was found between the knowledge score and factor 2, which implied that people who did not have access to piped water and those who used rainwater and water transported by tanker truck had a lower level of knowledge about schistosomiasis. On the other hand, the significant and positive association with factor 3 showed that interviewees who had already participated in some health intervention, who knew someone who had already had schistosomiasis and who had obtained information about the disease through a public health program possibly had better knowledge about the disease. Factors 5 and 6 showed a threshold very close to statistical significance, pointing out that possibly people who were informed about the disease by a doctor and/or at the hospital may signal better knowledge. However, women who were not economically active had a lower level of knowledge about parasitosis (Table 5).

Regarding the attitude score, it was proven that factors 1 and 3, respectively, positively impacted the population attitudes, so that people who lived in rural areas, were elderly, farmers, had a low level of education and were not informed about schistosomiasis in school had better attitudes. Individuals that had experience of participating in some public health program, knew someone who had schistosomiasis and were informed about the disease through a public health program shared better levels of attitudes (Table 5).

Still on attitudes, factors 4 and 7 had a significant association, although negative, showing that people who did not have piped water, used well water and found out about the disease from acquaintances and/or neighbors had worse attitudes compared to others. In addition, people who used the HC services and who had never had schistosomiasis also had unfavorable attitudes (Table 5).

Regarding the practice score, it showed a significant and positive association with factors 3 and 7, confirming that those who had participated in some public health program, knew someone who had schistosomiasis and had information about a public health program as a source had better practices, as well as people who used HC services and had never had schistosomiasis. Furthermore, people who did not have access to piped water, who used rainwater and water transported by tanker truck, as well as those who were informed about the disease by a doctor and/or at the hospital had unfavorable practices, according to a significant negative association of the practice score with factors 2 and 5 (Table 5).

## 4. Discussion

This research, conducted in an area endemic for schistosomiasis mansoni, showed that analyses using the KAP method were important to detect vulnerabilities that surround the knowledge, attitudes and practices of the population regarding this public health problem, that has been present in the Brazilian scenario for decades. It was identified that the attitude score was the highest in comparison to practices and knowledge, the latter having the lowest score. Such findings reinforce the importance of increasing health education to strengthen KAP in Brazilian communities. 

We evidenced that about a third of the individuals were not considered by the public water supply network, a fact that is extraordinarily remarkable, as water scarcity, as well as its quality for drinking, appeared as factors correlated to the occurrence of intestinal parasitosis [36]. On the other hand, when considering those who use piped water systems, a reduction in the scores related to practices can be seen, possibly due to the lack of knowledge regarding the severity of waterborne diseases on a daily basis. It is known that a fragile water supply in Brazilian regions, especially in the northeast, can increase the risk of exposure to diseases, such as schistosomiasis, when compared to regions with a structured and secure water supply [37]. 

According to the history of positive cases of schistosomiasis in the studied region and small participation in prevention campaigns, two important points were observed. The first refers to a positive point considering that most interviewees had never had schistosomiasis, even though they lived in a historically endemic area, which is corroborated by other researchers [38,39]. The second point refers to the fact that this happens even though most interviewees reported that they had never participated in schistosomiasis diagnosis and prevention campaigns through control program actions, differing from other reports [22,40].

The absence of health campaigns aimed at the prevention of schistosomiasis, verified by the remote participation of the analyzed public in intervention programs, becomes worrisome, considering the context of the endemic area in which the population resides, as well as the history of high positivity rates in the municipality, which makes them exposed to a risk situation and, under-reported cases, a fact that incurs the silent maintenance of the disease cycle.

The association findings among KAP components revealed that the level of attitudes stood out in relation to the practices and knowledge of the interviewees, something already pointed out in other similar studies [39,41,42,43]. This finding may suggest that the population’s perception of the seriousness of schistosomiasis is inherent in the knowledge.

Our findings pointed to a good level of prevention practices in the studied population, in contrast to lower levels of knowledge regarding schistosomiasis mansoni, involving forms of transmission, diagnosis, treatment, symptoms and prevention. Eventually, knowledge being inferior to attitudes and practices may be a reflection of a series of habits that were already existing in the population’s daily life and that were naturally repeated without people knowing exactly why. That is, people want to take care of themselves (higher attitude) but do not know exactly how to do it (lower knowledge), resulting in less health care (lower practice). This disparity between practices and knowledge was present in other countries, [18,38,39,40,41,43,44], highlighting that socioeconomic factors should also be considered, implying greater difficulty in self-care. Furthermore, it must be taken into account that the correct knowledge some people think they have about schistosomiasis is not supported by scientific evidence, but by socially shared knowledge, which can result in resistance to changes in behavior.

Of further concern is the fact that even though the population lives in an endemic area for schistosomiasis, it is common that people may never have heard of the disease [39] or know it by local popular names [18] through case reports that end up resulting in inconsistent information and knowledge [6,10,29,40].

In summary, our findings show a positive correlation between knowledge, attitudes and practices, where better levels of knowledge imply better attitudes and practices. With this, knowledge exerts influence on the construction of a healthier attitude, resulting in adherence to good behaviors, attitudes and practices in the search for health services and care that contribute to schistosomiasis prevention and control [44,45,46].

Factors associated with the best levels of all KAP components in schistosomiasis prevention were older age group, no provision of water through rainfall and previous history of the disease. The relationship between the sociodemographic profile and knowledge allows us to create intersections with other studies, which identified that older individuals know more about the disease [41], such as that observed in the present study.

In the attitudes component, as in knowledge, age also had significant relationships [47]. Therefore, it is possible to assume that older people may have had experiences that provide them with a more refined sense of self-care and, consequently, better attitudes. There was an association between gender and age with the component of self-care practices [38,40]. 

Furthermore, a positive case history for schistosomiasis was associated with the three KAP components, showing that people who have already had the infection at some point in their lives have better levels of knowledge, attitudes and prevention practices regarding the parasitic disease. Such results can be related to a professional orientation (in health services) regarding transmission, diagnosis, treatment and prevention. Furthermore, having participated in health actions promoted by the schistosomiasis control program, knowing someone who has already had the disease, and having been informed about it through a public health program were also factors associated with knowledge and practices. In line with this, the literature mentions that a previous diagnosis of schistosomiasis was associated with KAP [47], reflecting positive attitudes towards health care related to disease prevention [22,39].

Considering our findings, and the scientific literature, it is believed that as age and level of schooling advance, certainly, this can broaden access to more concrete information about schistosomiasis, which would justify its greater engagement and management in disease prevention [18,44,47].

The profile found in our study, with a predominance of female people, can be explained because women historically had greater health care in comparison to men, either for the active treatment of a given disease or preventive care [48,49], although there are studies that have identified that men know the disease better [18,47]. We reiterate that the greater health care observed in women [48,49], as well as the possible influence that life experiences can bring, contribute to a better sense of self-care and preventive practices.

The low percentage of interviewers (16.02%) who indicated school as a source of knowledge about schistosomiasis can be seen as worrisome, considering that the school environment is favorable to information propagation that can positively reflect in critical thinking and healthier life habits [50,51,52]. In some cases, having heard about the disease in the school environment was associated with the intention to have good health care attitudes [39]. The multivariate analysis of the KAP scores highlighted factor 3 as the one that seemed to have an important impact on the population’s KAP. Such a finding leads to the belief that participation and receiving information through health intervention programs like the SCP, even for those people who have not tested positive for schistosomiasis, can possibly open possibilities of access not only to the diagnosis and treatment of the disease but also to knowledge that can provide greater altruism in the population in self-care in health, as documented in the literature for those who have already had the disease [22,39,47].

In addition, it is worth mentioning that the interviewees who know someone who has already had schistosomiasis demonstrate good knowledge and practices, parallel to this variable, integrating factor 3. This factor’s influence on KAP possibly occurred due to the fact that people with a history of illness have been correctly guided by health professionals, so they were able to understand the knowledge that they were offered.

It is possible that the municipal SCP, in the scope of the actions and services offered, lacks individual and collective education activities with the municipality’s population, which may contribute to the worsening of the disease situation in Feira Grande/AL. Our findings support this hypothesis, as it was observed that people who had participated in some health intervention and had been informed about the disease through public health programs had better levels of KAP.

We reiterate the importance and contributions of control programs that perform health intervention actions for diagnosis, treatment [53] and education about schistosomiasis, enabling their awareness through access to adequate knowledge, good attitudes and the best preventive practices. We emphasize that actions such as health education can be developed through educational programs implemented in parallel or separately from SCP actions, positively impacting the population’s knowledge and adherence to positive attitudes in health care [54,55,56].

Some limitations were observed throughout the study. First, the COVID-19 pandemic may have impacted the field investigations. Second, the convenient choice of center health and the risk of bias associated with the use of community health agents to reach the intended sample and a possible memory bias by stimulating the retrieval of previous experiences and the knowledge of the participants. However, we emphasize that the aforementioned limitations in no way impacted or hindered the investigation and analysis of the data, as well as the interpretation of the results presented.

## 5. Conclusions

The variables of attitudes and knowledge had the highest and lowest scores, respectively. Age group, water supply source and history of past schistosomiasis were factors associated with KAP in the study population. Therefore, we emphasize that our findings contributed to broadening perceptions about schistosomiasis prevention and to deeply understanding the dynamics of the three components with socioeconomic factors and searching for the services and health care of the analyzed population. The present study highlights the impacts that programs and health interventions can possibly influence in controlling the disease.

Lower levels of the interviewees’ knowledge provided evidence of the need for health education actions focused on schistosomiasis aiming at improving knowledge and consequently prevention attitudes and practices, to help the population become more active in health care regarding schistosomiasis.

## Figures and Tables

**Table 1 tropicalmed-08-00034-t001:** Characteristics, sociodemographic and health care demand profiles of residents from Feira Grande, Alagoas, Brazil, in 2021.

Characteristics	N	%
**Residence area**		
Rural	292	70.87
Urban	88	21.36
Peri-urban	32	7.77
**Sex**		
Female	306	74.27
Male	106	25.73
**Age group**		
Youth (15 to 19)	16	3.88
Young Adult (20 to 39)	182	44.17
Adult (40 to 59)	125	30.34
Senior (60 or +)	89	21.60
**Education in years of study**		
Illiterate or Elementary I Incomplete	167	40.53
Elementary I Complete or Elementary II Incomplete	51	12.38
Elementary II Complete or Middle School Incomplete	56	13.59
High School Complete or Superior Incomplete	105	25.49
High School Complete	33	8.01
**Economically active**		
Yes	143	34.71
No	269	65.29
**Profession**		
Farmer (a)	268	65.05
Another	70	16.99
None	74	17.96
**Water supply sources**		
Piped water	293	71.12
Well water	78	18.93
Water from the cacimba	68	16.50
Rainwater	17	4.13
Weir water	1	0.24
Water transported by the cistern truck	6	1.46
**Has a bathroom in the residence**		
Yes	411	99.76
No	1	0.24
**Frequency you use the HC services**		
At least once per month	58	14.08
Every 3 months	9	2.18
Every 6 months	5	1.21
Only when necessary	340	82.52
**Self-reported history of past schistosomiasis**		
Yes	80	22.66
No	273	77.34
**Participated in some SCP action**		
Yes	183	47.29
No	204	52.71
**Do you know someone who had schistosomiasis**		
Yes	234	56.93
No	177	43.07
**Disease knowledge sources**		
Public Health Services	134	32.52
Doctor or hospital	42	10.19
School	66	16.02
Acquaintances and/or neighbors	93	22.57
Relatives and/or older persons	92	22.33
None	36	8.74
**Total**	412	100

Abbreviations: HC: Health Center; SCP: Schistosomiasis Control Program.

**Table 2 tropicalmed-08-00034-t002:** Scores of knowledge, attitudes and practices of residents from Feira Grande, Alagoas, Brazil, in 2021.

Scores	K	A	P	Md	IQR	Minimum	Maximum
	(*Spearman* Correlation)						
Knowledge (K)	1.000	0.351 *	0.301 *	0.66	0.32	0.00	0.98
Attitudes (A)	0.351 *	1.000	0.292 *	0.83	0.00	0.17	1.00
Practices (P)	0.301 *	0.292 *	1.000	0.67	0.17	0.00	1.00

Note: * Statistical significance at *p* < 0.05 by Spearman test. Abbreviations: Md, Median; IQR, Interquartile Interval.

**Table 3 tropicalmed-08-00034-t003:** Association of knowledge, attitude and practice scores with socio-demographic profile of residents from Feira Grande, Alagoas, Brazil, in 2021.

Characteristics	Knowledge			Attitude			Practices		
	Md (IQR)	*p*-Value	E.S.	Md (IQR)	*p*-Value	E.S.	Md (IQR)	*p*-Value	E.S.
**Residence area**		0.740			0.329			**0.020**	0.014 ^A^
Rural	0.66 (0.30)			0.83 (0.17)			0.67 (0.17)		
Urban	0.66 (0.35)			0.83 (0.00)			0.50 (0.17)		
Peri-urban	0.64 (0.42)			0.83 (0.00)			0.50 (0.17)		
Sex		0.961			0.232			**0.040**	0.101 ^A^
Female	0.66 (0.32)			0.83 (0.00)			0.67 (0.17)		
Male	0.65 (0.25)			0.83 (0.00)			0.50 (0.34)		
**Age Group**		**0.001**	0.034 ^A^		**0.000**	0.054 ^A^		**0.014**	0.019 ^A^
Youth (15 to 19)	0.45 (0.22)			0.75 (0.16)			0.50 (0.21)		
Young Adult (20 to 39)	0.63 (0.33)			0.83 (0.16)			0.50 (0.17)		
Adult (40 to 59)	0.67 (0.32)			0.83 (0.17)			0.67 (0.17)		
Senior (60 or +)	0.7 (0.27)			0.83 (0.17)			0.67 (0.17)		
**Education in years of study**		**0.005**	0.026 ^A^		0.065			0.396	
Illiterate or Elementary I Incomplete	0.66 (0.29)			0.83 (0.17)			0.67 (0.17)		
Elementary I Complete or Elem. II Incom.	0.60 (0.39)			0.83 (0.00)			0.67 (0.17)		
Elem. II Com. or Middle School Incom.	0.62 (0.35)			0.83 (0.00)			0.67 (0.17)		
High School Complete or Superior Incom.	0.63 (0.34)			0.83 (0.16)			0.50 (0.17)		
High School Complete	0.74 (0.16)			0.83 (0.00)			0.67 (0.17)		
**Economically active**		**0.002**	0.156 ^A^		0.285			0.727	
Yes	0.70 (0.29)			0.83 (0.00)			0.67 (0.17)		
No	0.63 (0.31)			0.83 (0.00)			0.67 (0.17)		
**Profession**		**0.012**	0.017 ^A^		0.115			0.233	
Farmer (a)	0.63 (0.31)			0.83 (0.17)			0.67 (0.17)		
Another	0.72 (0.27)			0.83 (0.00)			0.50 (0.17)		
None	0.68 (0.28)			0.83 (0.16)			0.50 (0.17)		
**Water supply sources**									
*Piped water*		**0.006**	0.136 ^A^		-			**0.000**	0.176 ^A^
Yes	0.66 (0.28)			0.83 (0.17)			0.50 (0.17)		
No	0.60 (0.37)			0.83 (0.16)			0.67 (0.17)		
*Well water*		0.723			-			0.128	
Yes	0.66 (0.31)			0.83 (0.16)			0.67 (0.17)		
No	0.66 (0.31)			0.83 (0.17)			0.50 (0.17)		
*Water from the cacimba*		0.804			0.734			0.073	
Yes	0.67 (0.33)			0.83 (0.00)			0.50 (0.17)		
No	0.66 (0.30)			0.83 (0.00)			0.67 (0.17)		
*Rainwater*		**0.005**	0.137 ^A^		**0.003**	0.146 ^A^		**0.000**	0.180 ^A^
Yes	0.41 (0.40)			0.67 (0.16)			0.33 (0.33)		
No	0.66 (0.30)			0.83 (0.00)			0.67 (0.17)		
*Water transported by the cistern truck*		0.531			0.265			**0.007**	0.132 ^A^
Yes	0.48 (0.33)			0.75 (0.16)			0.25 (0.29)		
No	0.66 (0.31)			0.83 (0.00)			0.67 (0.17)		

Note: Statistical significance at *p* < 0.05 by Mann–Whitney and Kruskal–Wallis tests. Abbreviations: Md, Median; IQR. Interquartile Interval; Elem, Elementary; Comp, Complete; Incom, Incomplete, -inconclusive results; E.S., Effect Size; ^A^ Small effect size.

**Table 4 tropicalmed-08-00034-t004:** Association of the scores of knowledge, attitudes and practices with the profile of health care demand and services among residents of Feira Grande, Alagoas, Brazil, in 2021.

Characteristics	Knowledge			Attitude			Practices		
	Md (IQR)	*p*-Value	E.S.	Md (IQR)	*p*-Value	E.S.	Md (IQR)	*p*-Value	E.S.
**Frequency you use the HC services**		0.516			0.215			-	
At least once per month	0.66 (0.31)			0.83 (0.29)			0.67 (0.17)		
Every 3 months	0.76 (0.18)			0.83 (0.00)			0.67 (0.00)		
Every 6 months	0.70 (0.16)			1.00 (0.17)			0.67 (0.16)		
Only when necessary	0.66 (0.31)			0.83 (0.00)			0.67 (0.17)		
**Self-reported history of past schistosomiasis**		**0.000**	0.202 ^A^		**0.000**	0.388 ^B^		**0.002**	0.163 ^A^
Yes	0.70 (0.26)			1.00 (0.17)			0.67 (0.17)		
No	0.63 (0.35)			0.83 (0.16)			0.50 (0.17)		
**Participated in some SCP action**		**0.000**	0.193 ^A^		-			**0.000**	0.296 ^A^
Yes	0.68 (0.27)			0.83 (0.17)			0.67 (0.17)		
No	0.58 (0.34)			0.83 (0.16)			0.50 (0.17)		
**Do you know someone who had schistosomiasis**		**0.000**	0.241 ^A^		-			**0.015**	0.120 ^A^
Yes	0.68 (0.28)			0.83 (0.17)			0.67 (0.17)		
No	0.56 (0.36)			0.83 (0.00)			0.50 (0.17)		
**Disease knowledge sources**									
*Public Health Services*		**0.000**	0.196 ^A^		-			**0.000**	0.209 ^A^
Yes	0.68 (0.25)			0.83 (0.17)			0.67 (0.17)		
No	0.61 (0.35)			0.83 (0.16)			0.50 (0.17)		
*Doctor or hospital*		**0.004**	0.140 ^A^		0.134			0.961	
Yes	0.72 (0.18)			0.83 (0.17)			0.67 (0.17)		
No	0.63 (0.33)			0.83 (0.00)			0.67 (0.17)		
*School*		0.051			0.760			0.140	
Yes	0.70 (0.28)			0.83 (0.00)			0.50 (0.17)		
No	0.63 (0.33)			0.83 (0.00)			0.67 (0.17)		
*Acquaintances and/or neighbors*		0.464			0.521			0.898	
Yes	0.63 (0.31)			0.83 (0.00)			0.67 (0.17)		
No	0.66 (0.31)			0.83 (0.00)			0.67 (0.17)		
*Relatives and/or older persons*		0.442			0.977			0.609	
Yes	0.63 (0.33)			0.83 (0.41)			0.67 (0.17)		
No	0.66 (0.30)			0.83 (0.00)			0.67 (0.17)		

Note: Statistical significance at *p* < 0.05 by Mann–Whitney and Kruskal–Wallis tests. Abbreviations: Md—Median; IQR—Interquartile Interval; (-) inconclusive results; E.S.—Effect Size; ^A^ Small effect size; ^B^ Moderate effect size.

**Table 5 tropicalmed-08-00034-t005:** Multiple associations of the scores of knowledge, attitudes and practices with the factors that influence the KAP of the residents from Feira Grande, Alagoas, Brazil in 2021.

Score	Variables	Estimate	Standard Error	95% CI	*p*-Value
**Knowledge**	**Intercept**	0.30	0.05	0.20; 0.41	**<0.001**
Pseudo R^2^ = 0.1291	**Factor 1**	−0.12	0.07	−0.25; 0.01	0.075
	**Factor 2**	−0.21	0.07	−0.35; −0.08	**0.002**
	**Factor 3**	0.51	0.07	0.39; 0.64	**<0.001**
	**Factor 4**	0.00	0.06	−0.11; 0.11	0.999
	**Factor 5**	0.11	0.06	0.00; 0.23	0.053
	**Factor 6**	−0.12	0.06	−0.23; 0.00	0.051
	**Factor 7**	−0.05	0.06	−0.16; 0.07	0.433
	**Factor 8**	0.00	0.05	−0.11; 0.11	0.962
**Attitude**	**Intercept**	1.78	0.08	1.63; 1.93	**<0.001**
Pseudo R^2^ = 0.1788	**Factor 1**	0.30	0.07	0.16; 0.44	**<0.001**
	**Factor 2**	−0.11	0.07	−0.25; 0.03	0.114
	**Factor 3**	0.49	0.07	0.36; 0.63	**<0.001**
	**Factor 4**	−0.27	0.06	−0.39; −0.15	**<0.001**
	**Factor 5**	0.08	0.06	−0.04; 0.21	0.173
	**Factor 6**	−0.05	0.06	−0.17; 0.08	0.454
	**Factor 7**	−0.16	0.06	−0.28; −0.04	**0.009**
	**Factor 8**	0.11	0.06	−0.01; 0.22	0.068
**Practices**	**Intercept**	0.25	0.05	0.16; 0.34	**<0.001**
Pseudo R^2^ = 0.1087	**Factor 1**	−0.01	0.06	−0.12; 0.10	0.878
	**Factor 2**	−0.31	0.06	−0.43; −0.19	**<0.001**
	**Factor 3**	0.26	0.05	0.15; 0.37	**<0.001**
	**Factor 4**	0.06	0.05	−0.04; 0.15	0.248
	**Factor 5**	−0.13	0.05	−0.22; −0.03	**0.011**
	**Factor 6**	0.04	0.05	−0.06; 0.14	0.425
	**Factor 7**	0.12	0.05	0.02; 0.21	**0.017**
	**Factor 8**	0.02	0.05	−0.07; 0.11	0.658

Note: Statistical significance at *p* < 0.05 by the beta regression model. Abbreviations: 95% CI-95% Confidence Interval.

## Data Availability

The dataset generated during the current study is not publicly available but is available from the corresponding author on reasonable request.

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
