# Peer review of "Knowledge, Attitude and Practices towards the Prevention of Schistosomiasis Mansoni in an Endemic Area of Alagoas, Northeast Brazil"

_tropicalmed, 2023, doi:10.3390/tropicalmed8010034_

Round 1

Reviewer 1 Report

The manuscript reports a cross-sectional study, developed in the city of Feira Grande, State of Alagoas (Brazil), involving 412 participants (calculated sample); analyzes Knowledge, Attitudes and Practices (KAP) in relation to schistosomiasis mansoni.

Comments:

1. There are a large number of editing errors, mainly due to the lack of spaces between words (example: pg7, line 214 "hadbetter", pg 7, line263 "variableshad")

2. On line 28, page 1, abstract, there is an error referring to the number of respondents from rural areas: 70.87% of 412=292 and not 192. This error is repeated on page 2, line 46; on page 5, line 203 and also in table 1.

3.Table 4, "practices". Change to "practices"

4. page 12, table 6 is incorrect: table 5 is correct.

5. I didn't understand the statement placed on page 13, line 372" We evidenced that almost half of the people were analyzed not contemplated by the 372

public water supply network, a fact that is extraordinarily remarkable..." If I'm not mistaken, 71.12% of respondents had a public water supply; that's well over half!

6. Is the statement that schistosomiasis is the most common waterborne disease in the Americas correct? Please recheck this information.

Author Response

Dear reviewer, thank you very much for your attention and for the suggestions given to our manuscript. Respecting your opinion, we seek to meet all your recommendations, as follows:

  1. The manuscript reports a cross-sectional study, developed in the city of Feira Grande, State of Alagoas (Brazil), involving 412 participants (calculated sample); analyzes Knowledge, Attitudes and Practices (KAP) in relation to schistosomiasis mansoni

1.1 There are a large number of editing errors, mainly due to the lack of spaces between words (example: pg7, line 214 "hadbetter", pg 7, line263 "variableshad")

-Authors answer - Thank you very much for this comment, it was a typo already corrected.

  1. On line 28, page 1, abstract, there is an error referring to the number of respondents from rural areas: 70.87% of 412=292 and not 192. This error is repeated on page 2, line 46; on page 5, line 203 and also in table 1.

-Authors answer - Thank you very much for this comment, it was a typo already corrected.

3.Table 4, "practices". Change to "practices"

-Authors answer - Thank you very much for this comment, it was a typo already corrected.

  1. page 12, table 6 is incorrect: table 5 is correct.

-Authors answer - Thank you very much for this comment, it was a typo already corrected.

  1. I didn't understand the statement placed on page 13, line 372" We evidenced that almost half of the people were analyzed not contemplated by the public water supply network, a fact that is extraordinarily remarkable..." If I'm not mistaken, 71.12% of respondents had a public water supply; that's well over half!

Authors answer - Thank you very much for this comment, we corrected the proper aspect ratio.

  1. Is the statement that schistosomiasis is the most common waterborne disease in the Americas correct? Please recheck this information.

-Authors answer – We used two current references that support this statement, however we prefer to assume that it is among the most common without necessarily pointing out the “ranking”.

References:

CDC, C. F. D. C. A. P. Parasites - Schistosomiasis. CENTER FOR DISEASE CONTROL AND PREVENTION (CDC). https://www.cdc.gov/parasites/schistosomiasis/ (accessed 2020-04-16).

Noya, O.; Katz, N.; Pointier, J.P.; Theron, A.; de Noya, B.A. Schistosomiasis in America. In Neglected Tropical Diseases - Latin America and the Caribbean; Springer, Vienna, 2015; pp 11–43. https://doi.org/10.1007/978-3-7091-1422-3_2.

Reviewer 2 Report

This manuscript describes the knowledge, attitude and practices towards the prevention of Schistosoma mansoni infection among 412 participants in an endemic area of Alagoas, Northeast Brazil. The study showed an overall low level of knowledge but high level of attitude. The study identified significant variables associated with the level of KAP in the studied population.

Overall, the aim of this study was not novel as it was well investigated by similar previous studies in different countries including Brazil. The manuscript has major limitations in the data analysis and findings presentation. Moreover, the standard of written English should be improved. Specific comments are given below.

MAJOR REVISION:

1.     The levels of KAP of the participants are largely unclear. The results in text and tables do not show the response of participants on the variables or items. What was the level of knowledge on transmission? Or source of infections? What was the knowledge on certain measures of prevention? What were the responses on attitude items? What were the practices related to prevention or risk of infection by these participants? The reader cannot find answers for these questions from the manuscript. It is also unclear the magnitude of significant variables on each component?

2.     The manuscript needs stylistic and English editing to correct several grammatical and typographical errors. Some language errors are highlighted on the manuscript file, attached with this report.

3.     The manuscript is too lengthy, particularly, discussion section that should be shortened substantially focusing on important findings and the important message to policy-makers. Association of good knowledge with good attitude and practices has been mentioned many times in the first 10 paragraphs of discussion section. This should be minimized. For example, this was repeated 4 times in a small paragraph (in lines 438, 439, 442 & 443). Moreover, over repetition of results should be avoided.

4.     Some calculations are incorrect and should be revised. Please refer to related minor comments below.

MINOR REVISION: (according to text flow)

5.     Change title to “Knowledge, attitude and practices towards the prevention of schistosomiasis mansoni in an endemic area of Alagoas, Northeast Brazil”.

6.     Line 28: “From all participants, 70.87% 28 (n=192) ..” But 192/412 = 46.6%.

7.     Line 29: 22.66% (n=80).  But 80/412 = 19.4%. All other calculations should be revised.

8.     Line 31: “having had schistosomiasis.” This can be changed to “history of past schistosomiasis”. Apply throughout the manuscript.

9.     Line 32: “Among the scores,” what scores?

10.  Line 35: remove “Thus,”

11.  Line 35 & 365: What does “KAP method” mean?

12.  Lines 35-37: Conclusion: this should be rewritten.

13.  Keywords are not suitable.

14.  Lines 48-49: “anemia, low ……” these are not symptoms.

15.  Does this paragraph include all schistosomiasis or only that caused by S. mansoni?

16.  Lines 51-52: This is a repetition of the sentence in lines 43-44.

17.  Line 59: “form of the disease” but the forms are not mentioned.

18.  Lines 59-61: cite references to support this statement.

19.  End of introduction: previous KAP studies done in Brazil should be mentioned in the last paragraph.

20.  Line 85: change to “This was a cross-sectional study ….”

21.  Lines 86-87: repetition.

22.  Line 124: it should be 384.

23.  Methods: all CH should be HC.

24.  Provide detailed description of the questionnaire, questions, responses, interviews.

25.  Line 203: but 192/412 = 46.6%

26.  Line 212: But 80/412 = 19.4%. Subsequent calculations should be revised.

27.  Table 2: no need to add symbols for abbreviations, e.g. Md and IQI. 

28.  Tables 2, 3 and 4: IQI “Interquartile Interval” or IQR Interquartile range should be of two values.

29.  Tables 3 and 4: It is totally unclear what the levels of KAP of the participants were. Provide number of participants for each question (variable) and for each component (K, A, P) in both tables. Columns for E.S. can be removed

30.  Univariate analysis was done with median scores while multivariate analysis was done using dichotomous variables

31.  All Tables: “Source: Prepared by the authors (2022).” Remove this from footnotes.

32.  Lines 369-370: “indicating that the population's level of knowledge about schistosomiasis is lower than the levels of attitudes and practices.” This can be removed.

33.  Line 446: what does “best levels of KAP” mean here?

34.  Use the abbreviation (KAP) throughout the manuscript. Apply first appearance rule.

35.  The variable of water sources is confusing. It is mentioned as not using rain water; and also had piped water and not using rain water; use water that came from the rain, and other words. For example, lines 250-251 “who had their houses supplied by the utility company and did not use rainwater”. This can be rephrased and standardized.

36.  Rain water or rainwater?

37.  Line 258: “who did not had piped water, did not used rainwater, or trans-258 ported by tanker truck,”. Please revise.

38.  Line 312: “did not have piped water and use water that came from the rain” please rephrase.

39.  Line 526: “and previous diagnosis of the disease” or previous history of the disease?

40.  Line 527-528 and also in abstract: ”we emphasize that the KAP method contributed to broaden perceptions about schistosomiasis prevention and …” this is really unclear.

41.  Line 531: “the positive impacts that programs and health interventions can have in controlling” this is inaccurate as no results to support this. “ can have on the population’s KAP”!

42.  Reference list should be prepared according to journal’s style.

Author Response

Revisor 2:

Dear reviewer, thank you very much for your attention and for the suggestions given to our manuscript. Respecting your opinion, we seek to meet all your recommendations, as follows:

MAJOR REVISION:

  1. The levels of KAP of the participants are largely unclear. The results in text and tables do not show the response of participants on the variables or items. What was the level of knowledge on transmission? Or source of infections? What was the knowledge on certain measures of prevention? What were the responses on attitude items? What were the practices related to prevention or risk of infection by these participants? The reader cannot find answers for these questions from the manuscript. It is also unclear the magnitude of significant variables on each component?

- Authors answer – We performed a descriptive analysis of the KAP, indicating frequency and percentage and inserted it as supplementary material.

  1. The manuscript needs stylistic and English editing to correct several grammatical and typographical errors. Some language errors are highlighted on the manuscript file, attached with this report.

-Authors answer – the manuscript was completely revised by a US company, which has native professionals.

  1. The manuscript is too lengthy, particularly, discussion section thatshould be shortened substantially focusing on important findings and the important message to policy-makers. Association of good knowledge with good attitude and practices has been mentioned many times in the first 10 paragraphs of discussion section. This should be minimized. For example, this was repeated 4 times in a small paragraph (in lines 438, 439, 442 & 443). Moreover, over repetition of results should be avoided.

- Authors answer – We thank you for your contribution. The text was revised again and the discussion was streamlined, by about 25%, it dropped from just over 2000 words to around 1500. Ideas were better organized and focused, and cohesion and coherence was improved.

  1. Some calculations are incorrect and should be revised. Please refer to related minor comments below.

- Authors answer - Revised and corrected calculations, thank you very much.

MINOR REVISION: (according to text flow)

  1. Change title to “Knowledge, attitude and practices towards the prevention of schistosomiasis mansoni in an endemic area of Alagoas, Northeast Brazil”.

-Authors answer – It has been corrected, thank you very much.

  1. Line 28: “From all participants, 70.87% 28 (n=192) ..” But 192/412 = 46.6%.

- Authors answer - It has been corrected, thank you very much.

  1. Line 29: 22.66% (n=80).  But 80/412 = 19.4%. All other calculations should be revised.

- Authors answer – Thank you very much for this comment. We have corrected the way in which the values are presented, since the total number of respondents for each item has changed, that is, not all questions were answered by the 412 participants.

  1. Line 31: “having had schistosomiasis.” This can be changed to “history of past schistosomiasis”. Apply throughout the manuscript.

- Authors answer - Thanks for your input, we've made the necessary adjustments.

  1. Line 32: “Among the scores,” what scores?

-Authors answer – It has been corrected

  1. Line 35: remove “Thus,”

-Authors answer – It has been corrected

  1. Line 35 & 365: What does “KAP method” mean?

-Authors answer – It has been corrected

  1. Lines 35-37: Conclusion: this should be rewritten.

-Authors answer – It has been corrected

  1. Keywords are not suitable.

-Authors answer – It has been corrected

  1. Lines 48-49: “anemia, low ……” these are not symptoms.

-Authors answer – It has been corrected

  1. Does this paragraph include all schistosomiasis or only that caused by S. mansoni?

-Authors answer – Exactly, the introduction was organized in order to bring the general details of the disease, and then present the specificities, and epidemiology of S. mansoni.

  1. Lines 51-52: This is a repetition of the sentence in lines 43-44.

-Authors answer – It has been corrected

  1. Line 59: “form of the disease” but the forms are not mentioned.

-Authors answer – It has been corrected

  1. Lines 59-61: cite references to support this statement.

-Authors answer – It has been corrected

  1. End of introduction: previous KAP studies done in Brazil should be mentioned in the last paragraph.

- Authors answer – There are no studies published in reliable literature, performed in Brazil using KAP and schistosomiasis mansoni. To reach this conclusion, we carried out a broad search with controlled descriptors in Portuguese, English and Spanish. However, if the reviewer has knowledge of texts in these conditions, please send us the references and we will be happy to include them.

  1. Line 85: change to “This was a cross-sectional study ….”

-Authors answer – It has been corrected

  1. Lines 86-87: repetition.

-Authors answer – It has been corrected

  1. Line 124: it should be 384.

- Authors answer - We redid the calculation and it is correct. We do not understand what parameters the reviewer used to arrive at this value. However, we are available for further clarification.

  1. Methods: all CH should be HC.

-Authors answer - It has been corrected

  1. Provide detailed description of the questionnaire, questions, responses, interviews.

-Authors answer – We added in the supplementary material a detail of the questions, and questionnaires, as well as the response percentages.

  1. Line 203: but 192/412 = 46.6%

-Authors answer - It has been corrected

  1. Line 212: But 80/412 = 19.4%. Subsequent calculations should be revised.

-Authors answer – In this case, the population does not correspond to 412, considering that there was a “missing” in this question, that is, not everyone responded. We remind you that according to the ethical legislation in Brazil, the participant is guaranteed not to answer all the questions.

  1. Table 2: no need to add symbols for abbreviations, e.g. Md and IQI. 

-Authors answer - Adjusted.

  1. Tables 2, 3 and 4: IQI “Interquartile Interval” or IQR Interquartile range should be of two values.

-Authors answer - Adjusted..

  1. Tables 3 and 4: It is totally unclear what the levels of KAP of the participants were. Provide number of participants for each question (variable) and for each component (K, A, P) in both tables. Columns for E.S. can be removed

-Authors answer – The percentages for each component are already detailed in the supplementary material. The article itself already covers a lot of content, bringing the descriptive percentage of each question will only inflate the results, since we will not have a viable space to discuss them, so we decided to place them in the supplementary material in respect of the objectives.

The hypothesis tests used are influenced by the sample size, in which it is more likely that the differences observed between the groups present a significant p-value. The effect size (E.S.) refers to a way to quantify the size of the difference between two groups, eliminating the effect of the sample size, complementing the significance of the test

The number of participants for each question is also detailed in Table 01, so repeating these questions would again “pollute” the results.

-30.  Univariate analysis was done with median scores while multivariate analysis was done using dichotomous variables

-Authors answer – As the scores did not present a Normal distribution (Kolmogorov-Smirnov test), comparisons between groups were made using non-parametric tests (Mann-Whitney and Kruskal-Wallis tests), and the median became the most appropriate statistic to compare the scores. groups. In the multivariate analysis, the scores became dependent variables in the model (Beta Regression Model), while the sociodemographic variables became independent. In order to be able to apply the Factor Analysis (to solve multicollinearity problems), the sociodemographic variables were binarized before applying the method.

  1. All Tables: “Source: Prepared by the authors (2022).” Remove this from footnotes.

-Authors answer – Adjusted

  1. Lines 369-370: “indicating that the population's level of knowledge about schistosomiasis is lower than the levels of attitudes and practices.” This can be removed.

-Authors answer – Adjusted

  1. Line 446: what does “best levels of KAP” mean here?

Authors answer - The KAP scores described within the closed interval between 0 and 1, where the higher the score value were, the higher the evaluation of knowledge, attitude and practices of the respondents. We point out that the mentioned justification is already described in the methodology of the work.

  1. Use the abbreviation (KAP) throughout the manuscript. Apply first appearance rule.

-Authors answer -  Adjusted

  1. The variable of water sources is confusing. It is mentioned as not using rain water; and also had piped water and not using rain water; use water that came from the rain, and other words. For example, lines 250-251 “who had their houses supplied by the utility company and did not use rainwater”. This can be rephrased and standardized.

-Authors answer - Considering the irregularity of the water supply service, it is possible for a person to use more than one supply method. Thus, a person who has piped water in his residence can also use rainwater in his daily life, for example. For this reason, the variables were treated as independent in the database.

  1. Rain water or rainwater?

-Authors answer - Rainwater.

  1. Line 258: “who did not had piped water, did not used rainwater, or trans-258 ported by tanker truck,”. Please revise.

-Authors answer - The variables in question correspond to different alternatives, where the respondent could select more than one alternative depending on their reality, that is, the person can have piped water and still use rainwater. It is important to point out that in Brazil, in addition to piped water supply, there are other forms that do not directly correspond to public supply, such as water from cisterns, wells, transported by water truck, among others, in this perspective, we seek to provide a greater universe of alternatives to the participants, aiming to cover their social reality.

  1. Line 312: “did not have piped water and use water that came from the rain” please rephrase.

-Authors answer - The variables in question correspond to different alternatives, where the respondent could select more than one alternative depending on their reality, that is, the person can have piped water and still use rainwater.

  1. Line 526: “and previous diagnosis of the disease” or previous history of the disease?

-Authors answer - We chose to adopt the history of past schistosomiasis, standardizing throughout the text. We appreciate the contribution.

  1. Line 527-528 and also in abstract: ”we emphasize that the KAP method contributed to broaden perceptions about schistosomiasis prevention and …” this is really unclear.

-Authors answer - Corrigido.

  1. Line 531: “the positive impacts that programs and health interventions can have in controlling” this is inaccurate as no results to support this. “ can have on the population’s KAP”!

-Authors answer - We emphasize that cross-sectional studies are not of cause and effect, a fact that does not allow us to assess causality...For this reason, we infer that there MAY possibly be a positive impact of public health programs since the participants who presented the best KAP score were those who had already underwent educational activities from public health programs.

  1. Reference list should be prepared according to journal’s style.

-Authors answer - Adjustments made.

Reviewer 3 Report

I reviewed manuscript (tropicalmed-2065594), “Knowledge, attitudes and practices (KAP) for the prevention of schistosomiasis mansoni in an endemic area of Alagoas, Northeast Brazil.”. This is a well-written paper on the results and analysis of the KAP method with questionnaires conducted in endemic area of schistosomiasis mansoni in Brazil. Points requiring correction are listed below.

1)    Throughout, the parasite name Schistosoma mansoni and the disease name schistosomiasis mansoni are mixed. Make sure to distinguish between the two.

2)    In P1L44, it states that schistosomiasis is an important waterborne disease in America, which I believe is incorrect. If this is the case, please cite the number of cases and other evidence.

3)    In P2L64, I do not express “carrier” in schistosomiasis. Does it mean antibody positive? Describe it clearly. Also, if possible, please indicate the recent infection rate of schistosomiasis and the rate of infection in the area you are studying. I also felt that the mention of schistosomiasis in introduction could use a more professional check. 

4)    In the method and beyond, the “HC” abbreviation for Health Center and the “CH” abbreviation are mixed. Unify them.

5)    In P3L129, informed Consent Form abbreviation (ICF) is doubled.

6)    In P3L143, submit the completed questionnaire as supplement data.

7)    The results are very well summarized and acceptable. However, there is no discussion as to why the practice scores were lower among those who use piped water systems.

8)    Discussion part is too long. Since you are explaining the same information over and over again, discuss what measures should be taken by breaking the discussion down by subject, such as age, education, water use, history of schistosomiasis, etc.

Author Response

Dear reviewer, thank you very much for your attention and for the suggestions given to our manuscript. Respecting your opinion, we seek to meet all your recommendations, as follows:

Revisor 3:

I reviewed manuscript (tropicalmed-2065594), “Knowledge, attitudes and practices (KAP) for the prevention of schistosomiasis mansoni in an endemic area of Alagoas, Northeast Brazil.”. This is a well-written paper on the results and analysis of the KAP method with questionnaires conducted in endemic area of schistosomiasis mansoni in Brazil. Points requiring correction are listed below.

1)    Throughout, the parasite name Schistosoma mansoni and the disease name schistosomiasis mansoni are mixed. Make sure to distinguish between the two.

Authors answer – We reviewed and made the necessary adjustments. We reiterate that according to the rules of scientific nomenclature, the name of the genus of the parasite begins with an initial capital letter, followed by the specific epithet in all lowercase and both words in italics: Schistosoma mansoni, distinguishing it from the name of the disease (Schistosomiasis mansoni), by the absence of such typographical resources, adopting capital initials in the name of the disease only when it begins a paragraph. Anyway, after the first mention of the names of the parasite and the disease, we opted for their abbreviations: S. mansoni and schistosomiasis, respectively, in the rest of the manuscript.

  1. In P1L44, it states that schistosomiasis is an important waterborne disease in America, which I believe is incorrect. If this is the case, please cite the number of cases and other evidence.

Authors answer – Adjusted

  1. In P2L64, I do not express “carrier” in schistosomiasis. Does it mean antibody positive? Describe it clearly. Also, if possible, please indicate the recent infection rate of schistosomiasis and the rate of infection in the area you are studying. I also felt that the mention of schistosomiasis in introduction could use a more professional check.

-Authors answer – The term “carrier” was used incorrectly, after reviewing the writing we used a more appropriate term. Regarding the recent infection rate of the disease in the studied area, we found and added relevant updated data referring only to the year 2021, possibly the impacts of the pandemic on the actions of the schistosomiasis control program, interfered in the notification of cases of the disease by the municipality studied . Anyway, thanks for the contributions.

  1. In the method and beyond, the “HC” abbreviation for Health Center and the “CH” abbreviation are mixed. Unify them.

-Authors answer – Adjusted.

5)    In P3L129, informed Consent Form abbreviation (ICF) is doubled.

-Authors answer – The second term was spelled wrong, we made the necessary correction.

6)    In P3L143, submit the completed questionnaire as supplement data.

-Authors answer – Suggestion accepted.

7)    The results are very well summarized and acceptable. However, there is no discussion as to why the practice scores were lower among those who use piped water systems.

-Authors answer – Rewritten.

8)    Discussion part is too long. Since you are explaining the same information over and over again, discuss what measures should be taken by breaking the discussion down by subject, such as age, education, water use, history of schistosomiasis, etc.

-Authors answer – We thank you for your contribution. The text was revised again and the discussion was streamlined, by about 25%, it dropped from just over 2000 words to around 1500. Ideas were better organized and focused, and cohesion and coherence was improved.

Reviewer 4 Report

Thank you for giving me the opportunity to review the article. The authors conducted a review focusing on the KAP for the prevention of schistosomiasis mansoni. I thought that the topic is socially important, but there are crucial methodological problems in the manuscript. Therefore, the reviewer thought the manuscript cannot be accepted for publication in the journal. I listed my comments below.

Comments:

1.      The authors should check typos before paper submission (e.g., use of space, mixed up of HC and CH).

Methods:

2.      There may be a selection bias. People who came to the HC are not general population, and the active search in residences can be affected by agents visiting plan.

3.      The authors used structured questionnaires with 12, 6 and 6 questions to assess the KAP. However, the validity of the questionnaire is unclear.

4.      The details of structured survey process are unclear.

5.      The details of statistical analysis with appropriate references are required.

Other sections:

6.      Why “Data Availability Statement” is “Not applicable”? The information is very important to improve the reliability of the study.

Author Response

Dear reviewer, thank you very much for your attention and for the suggestions given to our manuscript. Respecting your opinion, we seek to meet all your recommendations, as follows:

Revisor 4:

Comments and Suggestions for Authors

Thank you for giving me the opportunity to review the article. The authors conducted a review focusing on the KAP for the prevention of schistosomiasis mansoni. I thought that the topic is socially important, but there are crucial methodological problems in the manuscript. Therefore, the reviewer thought the manuscript cannot be accepted for publication in the journal. I listed my comments below.

Comments:

  1. The authors should check typos before paper submission (e.g., use of space, mixed up of HC and CH).

-Authors answer – Adjusted.

Methods:

  1. There may be a selection bias. People who came to the HC are not general population, and the active search in residences can be affected by agents visiting plan.

- Authors answer - We understand that the existence of selection bias is possible, which we consider a limitation of our study (added in the text, L000), however we used community agents as a strategy to reach the interviewees. In addition, we consider that in the municipality of Feira Grande there is coverage of the Family Health Strategy of 95%, a fact that minimizes the heterogeneity of the interviewed population.

  1. The authors used structured questionnaires with 12, 6 and 6 questions to assess the KAP. However, the validity of the questionnaire is unclear.
  2. The details of structured survey process are unclear.

- Authors answer – We used a questionnaire created by the authors themselves based on previous publications in peer-reviewed journals. The questionnaire was face-content validated by five PhD researchers with expertise in the subject using the Delphi technique, an efficient and consolidated methodology to generate consensus based on the opinion of professional experts in the subject. The questionnaire was made available to the group of researchers online and evaluated regarding the degree of importance of each question for the research object, taking into account objectivity, clarity and relevance. For this, a Likert-type scale was used (1 – very small, 2 – small, 3 – reasonable, 4 – large and 5 – very large). Two rounds were held until consensus was reached. The content validity coefficient (CVC) was used to analyze the agreement index, so that, to remain on the form, the question needed to reach a minimum percentage of 0.8 of agreement, percentage fulfilled by all items. Subsequently, the questionnaire was tested (pre-test) with 10 participants from the reference population, with no need to make changes.

References:

Sousa AFL, Hermann PRS, Fronteira I, Andrade D. Monitoring of postoperative complications in the home environment. Rev RENE. 2020;21:e43161. DOI: http://dx.doi.org/https://doi.org/10.15253/2175-6783.20202143161

Hernandez-Nieto RA. Contributions to statistical analysis. Merida: University of Los Andes; 2002

  1. The details of statistical analysis with appropriate references are required.

Authors answer – Adjusted

Other sections:

  1. Why “Data Availability Statement” is “Not applicable”? The information is very important to improve the reliability of the study.

Authors answer – Thank you very much for this comment. It was duly corrected;

Reviewer 5 Report

This manuscript entitled ‘’Knowledge, attitudes and practices (KAP) for the prevention of schistosomiasis mansoni in an endemic area of Alagoas, Northeast Brazil’’ presents a cross-sectional study conducted between March and May 2021, with 412 participants who live in the municipality of Feira Grande, Alagoas, Brazil It provides a brief overview of knowledge, attitudes and practices on the chistosomiasis that from all participants, 70.87% 28 (n=192) lived in rural areas, 22.66% (n=80) reported having had schistosomiasis mansoni and 52.71% 29 (n=204) never participated in schistosomiasis control program actions. The results of the article also show that public health program practices are important in controlling infectious communicable  diseases.  As a result, it is a well-organized, and well-written article (some minor revision are needed: Line 42; This parasitic disease insteda of This parastit disease; Line 76; the population regarding instead of the populationregarding).

Author Response

Dear reviewer, thank you very much for your attention and for the suggestions given to our manuscript. Respecting your opinion, we seek to meet all your recommendations, as follows:

Revisor 5:

Comments and Suggestions for Authors

This manuscript entitled ‘’Knowledge, attitudes and practices (KAP) for the prevention of schistosomiasis mansoni in an endemic area of Alagoas, Northeast Brazil’’ presents a cross-sectional study conducted between March and May 2021, with 412 participants who live in the municipality of Feira Grande, Alagoas, Brazil It provides a brief overview of knowledge, attitudes and practices on the chistosomiasis that from all participants, 70.87% 28 (n=192) lived in rural areas, 22.66% (n=80) reported having had schistosomiasis mansoni and 52.71% 29 (n=204) never participated in schistosomiasis control program actions. The results of the article also show that public health program practices are important in controlling infectious communicable  diseases.  As a result, it is a well-organized, and well-written article (some minor revision are needed: Line 42; This parasitic disease insteda of This parastit disease; Line 76; the population regarding instead of the populationregarding).

Authors answer - Thank you very much for your contribution. We inform you that we corrected it promptly.

Round 2

Reviewer 4 Report

Thank you for giving me the opportunity to review the revised version of this article. The authors corrected the manuscript partly, but there are fundamental problems which cannot be addressed. Therefore, the reviewer thought the manuscript cannot be accepted for publication in the journal. I listed my comments below.

Comments:

Methods:

1.      There may be a selection bias. People who came to the HC are not general population, and the active search in residences can be affected by agents visiting plan.

AR: We understand that the existence of selection bias is possible, which we consider a limitation of our study (added in the text, L000), however we used community agents as a strategy to reach the interviewees. In addition, we consider that in the municipality of Feira Grande there is coverage of the Family Health Strategy of 95%, a fact that minimizes the heterogeneity of the interviewed population.

AC: The Family Health Strategy only provide the access to the primary care. The coverage rate does not minimize the heterogeneity of the interviewed population.

2.      The authors used structured questionnaires with 12, 6 and 6 questions to assess the KAP. However, the validity of the questionnaire is unclear.

AR: We used a questionnaire created by the authors themselves based on previous publications in peer-reviewed journals. The questionnaire was face-content validated by five PhD researchers with expertise in the subject using the Delphi technique, an efficient and consolidated methodology to generate consensus based on the opinion of professional experts in the subject. The questionnaire was made available to the group of researchers online and evaluated regarding the degree of importance of each question for the research object, taking into account objectivity, clarity and relevance. For this, a Likert-type scale was used (1 – very small, 2 – small, 3 – reasonable, 4 – large and 5 – very large). Two rounds were held until consensus was reached. The content validity coefficient (CVC) was used to analyze the agreement index, so that, to remain on the form, the question needed to reach a minimum percentage of 0.8 of agreement, percentage fulfilled by all items. Subsequently, the questionnaire was tested (pre-test) with 10 participants from the reference population, with no need to make changes.

AC: The authors should explain the process in the main text of the manuscript. Not only that, why did the authors provide the results obtained from the questionnaire as the Supplementally Materials?

3.      The details of structured survey process are unclear.

AR:

4.      The details of statistical analysis with appropriate references are required.

AR: Adjusted.

AC: The authors should provide the point-by-point response.

Discussion:

5.      AC: The authors stated that “we emphasize that the aforementioned limitations in no way impacted or hindered the investigation and analysis of the data, as well as the interpretation of the results presented”, but the limitations can have a significant impact on the results.

Supplementary Materials:

6.      AC: The statement and the contents are not matched. Not only that, why did the authors not include this “main” results in the main text?